# Thermodynamics, statistical mechanics and the vanishing pore width limit of confined fluids

W. Dong [1,2✉], T. Franosch[3✉] & R. Schilling[4✉]

Temperature, particle number and volume are the independent variables of the Helmholtz free energy for a bulk fluid. For a fluid confined in a slit pore between two walls, they are usually complemented by the surface area. However, an alternative choice is possible with the volume replaced by the pore width. Although the formulations with such two sets of independent variables are different, we show they are equivalent and present their relations. Corresponding general statistical-mechanics results are also presented. When the pore width becomes very small, the system behaves rather like a two-dimensional (2D) fluid and one can wonder if thermodynamics still holds. We find it remains valid even in the limit of vanishing pore width and show how to treat the divergences in the normal pressure and the chemical potential so that the corresponding 2D results can be obtained. Thus, we show that the Gibbs surface thermodynamics is perfectly capable of describing small systems.

[1] Laboratoire de Chimie, UMR 5182 CNRS, Ecole Normale Supérieure de Lyon, 46, Allée d'Italie, 69364 Lyon Cedex 07, France. [2] State Key Laboratory of Chem/Biosensing and Chemometrics, College of Chemistry and Chemical Engineering, Hunan University, 410082 Changsha, China. [3] Institut für Theoretische Physik, Universität Innsbruck, Technikerstraße, 21A, A-6020 Innsbruck, Austria. [4] Institut für Physik, Johannes Gutenberg Universität Mainz, Staudinger Weg 9, 55099 Mainz, Germany. ✉email: wei.dong@ens-lyon.fr; thomas.franosch@uibk.ac.at; rschill@uni-mainz.de

Despite its phenomenological character, thermodynamics is a powerful framework providing universal relations between various thermodynamic functions, e.g., the equations of state. It is remarkable that thermodynamic potentials depend only on a quite small number of independent variables. For instance, for a one-component fluid in a box, the fluid-particle number, $N$, and the box volume, $V$, are such variables. With different choices of independent variables, various thermodynamic potentials can be obtained using Legendre transforms.

Thermodynamics was initially developed for homogeneous macroscopic systems. In such cases, extensive thermodynamic variables scale with the system size, e.g., its volume, $V$, while the conjugate intensive variables are independent of the system size. Statistical mechanics provides a microscopic justification of the macroscopic thermodynamics in the so-called thermodynamic limit, e.g., $V \to \infty$ and $N \to \infty$, keeping the particle density $\rho = N/V$ fixed. Macroscopic thermodynamics holds under the condition, $\lim_{V \to \infty} (\mathcal{A}/V) = 0$ where $\mathcal{A}$ is the total surface area of system. J. W. Gibbs proposed surface thermodynamics to go beyond the macroscopic one by including the surface contribution, proportional to $\mathcal{A}$, into thermodynamic potentials[1]. However, it is not clear whether Gibb's formulation is the unique way for elaborating the surface thermodynamics. Our work here shows that alternative variants are possible.

An important class of inhomogeneous systems in which surfaces play a salient role are confined fluids. For simplicity, we consider here a one-component classical fluid confined in a slit pore composed of two parallel and flat impenetrable walls (see Fig. 1). The area of a single wall is $A$ and the slit width is $H$. Since we will illustrate the thermodynamic and statistical-mechanics results for a colloidal liquid of monodisperse hard spheres (HS) of diameter, $\sigma$, the accessible width for the centers of hard spheres is $L = H - \sigma$ while for point particles, $L = H$ and the volume given by $V = LA$. Thermodynamic quantities of experimental interest are, e.g., the normal pressure and the surface tension resulting, respectively, from the change of pore width and wall area.

As pointed out above, Legendre transforms allow generating various thermodynamic potentials, thus leading to different thermodynamic formulations. However, it is much less well recognized that even with a same thermodynamic potential, multiple choices of independent variables exist in some situations, which result in different intensive variables and different relations concerning the intensive variables, e.g., Gibbs-Duhem equation. The motivation for working out these equivalent formalisms is also to offer a most suitable description for different experimental or simulation situations. Fluids confined to a slit-pore provide an illustrating example and we will show that even for one and the same thermodynamic potential, e.g., the Helmholtz free energy, two alternative thermodynamic formulations are possible. Besides the well-known Gibbs

formulation with the choice of $T$, $V$, $\mathcal{A} = 2A$, and $N$ as independent variables, i.e., considering $F(T, V, \mathcal{A}, N)$, there is an alternative one based on the choice of $T$, $L = V/A$, $A$ and $N$ as independent variables, considering $\bar{F}(T, L, A, N)$. Our first objective in the present work is to prove first that the alternative formulation is completely equivalent to Gibbs' one and second to find relations between them. The second one is to establish the microscopic statistical-mechanics description corresponding to the proposed thermodynamic formalism.

Intuitively, one expects that the 3D confined fluid should be able to transform to a 2D one in the limit of vanishing pore width, i.e., $L \to 0$. Nevertheless, the theoretical demonstration of this appealing idea does not appear to be a trivial task. There are several evident difficulties. First, the normal pressure and the chemical potential of the 3D confined fluid diverge in the vanishing pore-width limit. So, it is obvious that the thermodynamic functions of the 3D confined fluid do not transform directly to those of the 2D fluid when the limit of vanishing pore width is taken. Moreover, in the limit $L \to 0$, the system size in this direction becomes vanishingly small. One can seriously question the validity of thermodynamics (a theory for macroscopic systems) under this condition. T. L. Hill was the pioneer who tried to extend thermodynamics for small systems[2,3]. His approach, named now as nanothermodynamics[4–6], is attracting much renewed interest[7–14] (see the monograph of Bedeaux, Kjelstrup and Schnell[7] for a recent review). Is it necessary to resort to Hill's nanothermodynamics to study the dimensional cross-over problem described above? If yes, how can it be applied concretely, i.e., how to construct the replicas needed for Hill's nanothermodynamics? How can the divergences of the pressure and the chemical potential of the 3D confined fluid be coped with properly? What are the precise relations between the thermodynamic functions of the 3D and 2D systems? Searching the answers to these open questions constitutes the third objective of the present work.

We show the most salient difference between the alternative formulations based on $F(T, V, \mathcal{A}, N)$ or $\bar{F}(T, L, A, N)$ is that the respective intensive variable conjugated with the surface area is different, for the former it is the well-known surface tension while for the latter it is the averaged transverse pressure. It is revealed also that if the singularities in the normal pressure and in the chemical potential are treated properly in the limit of vanishing pore width, thermodynamics holds even in this limit and the 3D to 2D crossover can be achieved.

## Results and discussion
### Thermodynamics
*General thermodynamic relations.* Depending on the choice of independent variables, there exist multiple but equivalent formulations of thermodynamics. This is why different thermodynamic potentials, such as the Helmholtz or the Gibbs free energy, and the grand potential, etc. have been introduced. Nevertheless, it is less well recognized that even with a given thermodynamic potential, different choices of independent variables are also possible and lead to different thermodynamic formulations. Failure in treating properly such situations can lead to confusions. In order to illustrate this point, we will consider a fluid confined in a slit pore composed of two parallel impenetrable walls with accessible width, $L$, as shown in Fig. 1. It is to note that all the results in the present and the next subsections hold for any one-component fluid and any pore width. The well-known surface thermodynamics formulated by Gibbs chooses the volume, $V$, total surface area, $\mathcal{A}$, number of particles, $N$, and temperature, $T$, as independent variables, the Helmholtz free energy, $F(T, V, \mathcal{A}, N)$, as thermodynamic potential, which is

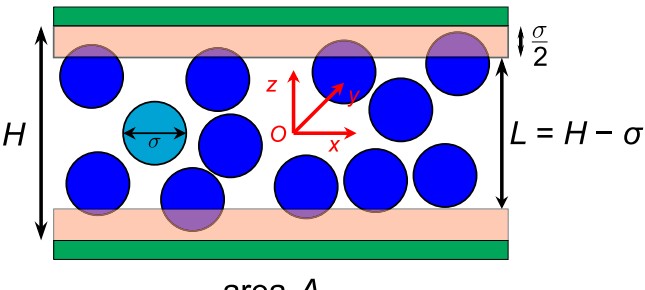

**Fig. 1 Schematic representation of a fluid of hard spheres confined in a slit pore formed by two hard walls.** Hard sphere (blue) diameter: $\sigma$; Pore width: $H$; Accessible pore width: $L = H - \sigma$; Surface area of one wall: $A$; Pore walls: green; Inaccessible region of hard sphere center: light brown.

described by the following fundamental equation,

$$dF = -SdT - p_\perp dV + \gamma d\mathcal{A} + \mu dN, \tag{1}$$

where the conjugated variables, i.e., pressure, $p_\perp$, surface tension, $\gamma$, chemical potential, $\mu$, and entropy, $S$, are defined respectively by,

$$p_\perp = -\left(\frac{\partial F}{\partial V}\right)_{T,\mathcal{A},N}, \tag{2}$$

$$\gamma = \left(\frac{\partial F}{\partial \mathcal{A}}\right)_{T,V,N}, \tag{3}$$

$$\mu = \left(\frac{\partial F}{\partial N}\right)_{T,V,\mathcal{A}}, \tag{4}$$

$$S = -\left(\frac{\partial F}{\partial T}\right)_{V,\mathcal{A},N}. \tag{5}$$

Due to the system's inhomogeneity, the surface region becomes also anisotropic. The obvious manifestation of the anisotropy is that near the surface, the pressure is no longer the same in different directions with respect to the surface normal. For the case of a flat surface considered here, the intensive variable conjugated to the volume is in fact the pressure perpendicular to the surface and this will be shown below more clearly and the index of $p_\perp$ is to indicate this explicitly. The fundamental equation, Eq. (1), describes a closed system in contact only with a thermal bath to exchange heat. For a slit pore of slab shape with the two walls of square shape with area, $A$, perpendicular to the z-axis (see Fig. 1), the volume and the total surface area are given respectively by,

$$V = AL, \tag{6}$$

$$\mathcal{A} = 2A + 4L\sqrt{A}. \tag{7}$$

Recall that $L = H$ for point particles. Since we will consider later the thermodynamic limit, $A \to \infty$ and $N \to \infty$ such that the 2D density, $n = N/A$, is fixed, the second term on the right-hand-side (RHS) of Eq. (7) is negligible. Consequently, $\mathcal{A} = 2A$ in our following discussions. As pointed out above, $p_\perp$ is in fact the normal pressure and this can be seen clearly by rewriting Eq. (2) as,

$$p_\perp = -\left(\frac{\partial F}{\partial V}\right)_{T,\mathcal{A},N} = -\frac{1}{A}\left(\frac{\partial F}{\partial L}\right)_{T,\mathcal{A},N}. \tag{8}$$

where $-(\partial F/\partial L)_{T,\mathcal{A},N}$ is nothing else but the force perpendicular to the pore walls.

We can also make an alternative choice of thermodynamic variables with the pore width, $L$, as independent variable instead of $V$. In addition, we will use $A = \mathcal{A}/2$ instead of $\mathcal{A}$. Note that $L$ and $A$ are the natural variables in a statistical-mechanics approach[15,16]. It is to be pointed out that this choice of independent variables for describing the Helmholtz free energy was used by F. Varnik[17] to devise a simulation method at constant normal pressure. Here, we present a systematic presentation of the formulation. Now, the fundamental equation for the free energy with such a choice of independent variables, i.e., $\bar{F}(T, L, A, N)$, becomes,

$$d\bar{F} = -\bar{S}dT - \bar{\Upsilon}dL - \bar{\Sigma}dA + \bar{\mu}dN, \tag{9}$$

where the connection between both free energies is given by

$$\bar{F}(T, L, A, N) = F(T, V = AL, \mathcal{A} = 2A, N), \tag{10}$$

and the other thermodynamic variables are,

$$\bar{S} = -\left(\frac{\partial \bar{F}}{\partial T}\right)_{L,A,N}, \tag{11}$$

$$\bar{\Upsilon} = -\left(\frac{\partial \bar{F}}{\partial L}\right)_{T,A,N}, \tag{12}$$

$$\bar{\Sigma} = -\left(\frac{\partial \bar{F}}{\partial A}\right)_{T,L,N}, \tag{13}$$

$$\bar{\mu} = \left(\frac{\partial \bar{F}}{\partial N}\right)_{T,L,A}. \tag{14}$$

Similar to $F$ and $\bar{F}$, $S$ and $\bar{S}$, as well as $\mu$ and $\bar{\mu}$, are different functions, although they are related directly via $S(T, V = AL, \mathcal{A}, N) = \bar{S}(T, L, A, N)$ and $\mu(T, V = AL, \mathcal{A}, N) = \bar{\mu}(T, L, A, N)$. From Eq. (8), we obtain readily,

$$\bar{\Upsilon} = -\left(\frac{\partial \bar{F}}{\partial L}\right)_{T,A,N} = p_\perp A. \tag{15}$$

$\bar{\Sigma}$ defined by Eq. (13) has the same physical dimension as the surface tension, $\gamma$, but it does not correspond to the well-known surface tension defined by Eq. (3) as it will be shown below. It is useful to note that $\bar{\Sigma}$ has also the same physical dimension as the pressure in a 2D system and that $\bar{\Sigma}/L$ has the dimension of the pressure in a 3D system. To clarify the physical meaning of $\bar{\Sigma}$, we rewrite Eq. (9) as,

$$d\bar{F} = -SdT - p_\perp AdL - \frac{\bar{\Sigma}}{L}LdA + \mu dN. \tag{16}$$

The second and third terms on the RHS of Eq. (16) describe respectively the work done by a volume change resulting from modifying the pore width, $L$, or pore surface area, $A$. For an isotropic bulk system, $p_\perp = \bar{\Sigma}/L = p$, the sum of these two terms reduces to $-pdV$ and we recover the well-known equation for a bulk system from Eq. (16). However, for a confined fluid interacting with the pore walls, the system becomes anisotropic, i.e., $p_\perp \neq p$ and the pressure in the pore is no longer a scalar but a tensor with unequal normal and transverse components[18]. Rearranging Eq. (16) as follows allows for clarifying the physical meaning of $\bar{\Sigma}$ and its relation to the surface tension,

$$d\bar{F} = -SdT - p_\perp dV + \frac{1}{2}\left(p_\perp - \frac{\bar{\Sigma}}{L}\right)Ld\mathcal{A} + \mu dN, \tag{17}$$

where $V = AL$ and $\mathcal{A} = 2A$ were used. Comparing Eq. (17) with Eq. (1), we obtain the thermodynamic relation between the surface tension, $\gamma$, and $\bar{\Sigma}$:

$$\gamma = \frac{L}{2}\left(p_\perp - \frac{\bar{\Sigma}}{L}\right). \tag{18}$$

The well-known mechanical definition of surface tension for a slit geometry[19] is given by,

$$\gamma = \frac{1}{2}\int_{-L/2}^{L/2} dz\left[p_\perp - p_\parallel(z)\right], \tag{19}$$

where $p_\perp$ and $p_\parallel$ are the normal and transverse components of the pressure tensor. Although $p_\perp$ is the same at any point in the pore to assure the mechanical equilibrium, the parallel component of the pressure tensor, $p_\parallel(z)$, varies with the position in the neighborhood of pore surfaces. From Eqs. (18) and (19), we

obtain immediately,

$$\bar{\Sigma} = \int_{-L/2}^{L/2} dz\, p_\parallel(z). \tag{20}$$

Hence, it is more appropriate to name $\bar{\Sigma}$ as integrated transverse pressure although it was called surface tension in the previous work of Franosch, Lang and Schilling[15]. In the context of Langmuir films, this is also referred as surface pressure. The expression for the pressure tensor found initially by Irving and Kirkwood is based on a mechanical definition and a particular choice of integration path to calculate the contribution of inter-particle interaction[18]. Schofield and Henderson pointed out that the choice of integration path is not unique[20]. The non-uniqueness for the mechanical definition of the pressure tensor have given rise to many debates in the literature (see, e.g., a recent study[21] and the references therein). Equations (18) and (19) show that to calculate the surface tension, one needs only the integrated normal and traverse components of the pressure tensor. The results given in Eqs. (13) and (15) provide a thermodynamic route for obtaining these integrated components from the derivative of the Helmholtz free energy and no choice of an integration path is needed for this route. The statistical-mechanics expressions obtained from this route will be presented below in Section III.

It is worthwhile to discuss also further consequences resulting from the different formulations above. For fluids confined to a slit pore, the free energy is a first-order homogeneous function only if all the extensive variables scale with the surface area, i.e.,

$$F(T, \lambda V, \lambda\mathcal{A}, \lambda N) = \lambda F(T, V, \mathcal{A}, N), \lambda > 0. \tag{21}$$

This leads immediately to the Euler relation,

$$F(T, V, \mathcal{A}, N) = -p_\perp V + \gamma\mathcal{A} + \mu N, \tag{22}$$

and the corresponding Gibbs-Duhem equation,

$$SdT - Vdp_\perp + \mathcal{A}d\gamma + Nd\mu = 0. \tag{23}$$

Alternatively, $\bar{F}$ exhibits the following scaling,

$$\bar{F}(T, L, \lambda A, \lambda N) = \lambda\bar{F}(T, L, A, N), \lambda > 0, \tag{24}$$

which leads to the Euler relation,

$$\bar{F}(T, L, A, N) = -\bar{\Sigma}A + \bar{\mu}N. \tag{25}$$

It is to note that as already pointed out earlier, $F(T, V, \mathcal{A}, N)$ and $\bar{F}(T, L, A, N)$, respectively, on the left-hand-side (LHS) of Eqs. (22) and (25) describe the same free energy [see Eq. (10)]. Due to the different choice of independent variables, the right-hand-sides (RHS) of Eqs. (22) and (25) look different but the results to be given in Sec. IV allow for showing that the sums give the same result. Taking the total differential of both sides of Eq. (25) and comparing the result with Eq. (16), we obtain the following Gibbs-Duhem-like relation for the alternative formulation with $\bar{F}$,

$$\bar{S}dT + A(p_\perp dL - d\bar{\Sigma}) + Nd\bar{\mu} = 0. \tag{26}$$

Although this relation does not look exactly the same as the Gibbs-Duhem equation in Eq. (23), they are closely related. Substituting $\bar{S} = S$, $d\bar{\mu} = d\mu$ and the identity, $p_\perp dL = d(p_\perp L) - Ldp_\perp$, into Eq. (26) and using the relation Eq. (18), we recover readily the Gibbs-Duhem equation given in Eq. (23).

Both formulations presented above contain the whole thermodynamic information, including that for phase transitions. When a phase transition takes place, both free energies become singular at, e.g., a critical temperature, $T_c$. Using $F(T, V, \mathcal{A}, N)$ or $\bar{F}(T, L, A, N)$ allows for investigating respectively the variation of $T_c$ with the volume or with the pore width.

We emphasize that the above thermodynamic formalism is valid for any value of the pore width. An extreme situation, i.e., a vanishing pore width, will be discussed in the next subsection. When the pore becomes narrower and narrower, some characteristic behaviors of small systems manifest themselves more and more. For example, differential and integral pressures ($p$, and $\hat{p}$), differential and integral surface tensions ($\gamma$ and $\hat{\gamma}$), differential and integral chemical potentials ($\mu$ and $\hat{\mu}$) are no longer the same[9,13,14]. The definitions of these differential and integral thermodynamic functions are recalled: $p = -(\partial F/\partial V)_{T,\mathcal{A},N} = -(\partial\Omega/\partial V)_{T,\mathcal{A},\mu}$ ($\Omega$: grand potential), $\hat{p} = -\Omega/V$, $\gamma = (\partial F/\partial\mathcal{A})_{T,V,N} = (\partial\Omega/\partial\mathcal{A})_{T,V,\mu}$, $\hat{\gamma} = (\Omega - \Omega^{bulk})/\mathcal{A}$, $\mu = (\partial F/\partial N)_{T,V,\mathcal{A}} = (\partial G/\partial N)_{T,V,\mathcal{A}}$, $\hat{\mu} = G/N$. T. L. Hill has been a pioneer for developing a thermodynamic approach for small systems[2–6]. Recently, W. Dong has shown that if the surface contribution is adequately accounted for, an alternative approach is possible, which is based on the more traditional surface thermodynamics[13,14]. It is to be emphasized that the thermodynamic formalism presented in this section applies for any arbitrarily small pore width. Moreover, we know now that the differential intensive variables are ensemble-independent while the integral intensive variables depend on ensembles[14]. In the present work, we consider only the differential intensive thermodynamic functions.

*Vanishing pore-width limit, $L \to 0$.* Although the expectation that the vanishing pore-width limit of a 3D confined fluid should lead to a 2D fluid is physically appealing, such a limit cannot be taken straightforwardly since certain singularities arise, e.g., the 3D density, $\rho = N/V$, diverges as $L^{-1}$, and the normal pressure diverges in the same way. Moreover, the chemical potential contains a logarithmic singularity such as $\ln L$. Nevertheless, these are removable singularities and we can deal with them properly by rewriting Eq. (22) as follows,

$$F + k_B TN\ln\frac{L}{\Lambda} = -(p_\perp L - 2\gamma)A + \left(\mu + k_B T\ln\frac{L}{\Lambda}\right)N, \tag{27}$$

where $k_B$ is the Boltzmann constant and $\Lambda$ the thermal wavelength. The terms, $k_B TN\ln(\Lambda L^{-1})$, on both sides of Eq. (27) is the difference between the Helmholtz free energy of an ideal gas in three and two dimensions. Now, the limit, $L \to 0$, can be taken and in this limit, Eq. (27) becomes the Euler equation of the 2D fluid,

$$F^{2D} = -p^{2D}A + \mu^{2D}N, \tag{28}$$

where

$$F^{2D} = \lim_{L\to 0}\left(F + k_B TN\ln\frac{L}{\Lambda}\right), \tag{29}$$

$$p^{2D} = \lim_{L\to 0}(p_\perp L - 2\gamma), \tag{30}$$

$$\mu^{2D} = \lim_{L\to 0}\left(\mu + k_B T\ln\frac{L}{\Lambda}\right). \tag{31}$$

Equations (29)–(31) give, respectively, the transformation of different thermodynamic quantities of the 3D confined fluid to those of a 2D fluid in the limit of vanishing slit width. Moreover, the Euler relation given in Eq. (28) shows clearly that the obtained 2D fluid is a bulk one since the free energy does not contain any interface contribution (nota bene: an interface in a 2D system is a line).

A similar treatment can be made for $\bar{F}$ given by Eq. (25). We find the Euler relation for the free energy of the 2D fluid,

$$\bar{F} + k_{\mathrm{B}}TN\ln\frac{L}{\Lambda} = -\Sigma A + \left(\mu + k_{\mathrm{B}}T\ln\frac{L}{\Lambda}\right)N. \qquad (32)$$

Now, taking the limit of vanishing slit width, we obtain,

$$\bar{F}^{2\mathrm{D}} = -\bar{p}^{2\mathrm{D}}A + \bar{\mu}^{2\mathrm{D}}N, \qquad (33)$$

where

$$\bar{F}^{2\mathrm{D}} = \lim_{L \to 0}\left(\bar{F} + k_{\mathrm{B}}TN\ln\frac{L}{\Lambda}\right), \qquad (34)$$

$$\bar{p}^{2\mathrm{D}} = \lim_{L \to 0}\bar{\Sigma} = \lim_{L \to 0}(p_{\perp}L - 2\gamma) = p^{2\mathrm{D}}, \qquad (35)$$

$$\bar{\mu}^{2\mathrm{D}} = \lim_{L \to 0}\left(\bar{\mu} + kT\ln\frac{L}{\Lambda}\right) = \lim_{L \to 0}\left(\mu + k_{B}T\ln\frac{L}{\Lambda}\right) = \mu^{2\mathrm{D}}. \qquad (36)$$

When going to the second equality of Eq. (35), we relied on Eq. (18) and $\bar{\mu}(T, L, A, N) = \mu(T, V = AL, \mathcal{A}, N)$ was used when going to the second equality of Eq. (36). Now, we see that $\bar{\Sigma}$ leads directly to the 2D fluid pressure in the limit of vanishing pore-width while it is the combination of normal pressure and surface tension in Eq. (30) which leads to the 2D fluid pressure. Although $F(T, V, \mathcal{A}, N)$ and $\bar{F}(T, L, A, N)$ are not the same function for a 3D confined fluid, the results of Eqs. (28)–(36) show that the functions $F^{2\mathrm{D}}$ and $\bar{F}^{2\mathrm{D}}$, depending on the same set of variables, i.e., $(T, A, N)$, are identical.

The logarithmic singularity arising in the chemical potential and the Helmholtz free energy in the limit $L \to 0$ is $\ln(L/\Lambda)$ if the particles interact with the pore surfaces via a hard-wall potential, $v_{HW}(z)$, (see Section IV). In more general cases described by a particle-wall interaction potential $v_W(z)$, the singularity becomes $\ln\left[\Lambda^{-1}\int_{-L/2}^{L/2}dz\exp\left\{-\beta\left[v_W(z + L/2) + v_W(L/2 - z)\right]\right\}\right]$ where $\beta = (k_BT)^{-1}$. In these cases, the proper 2D limit can be obtained similarly by removing such a singularity.

In the limit $L \to 0$, the size of the considered system becomes vanishingly small. Despite the fact that the 3D to 2D crossover in this limit is a physically appealing idea, it is by no means evident that thermodynamics (widely recognized as a theory for macroscopic system) is still valid in this limit. Hill's nanothermodynamics[2–6] is currently the best-known approach for describing the thermodynamic behavior of small systems. However, it does not seem possible to describe the 3D to 2D crossover by applying Hill's theory since even the surface tension does not enter explicitly into his theory. As already pointed out at the end of the last subsection, the thermodynamic formalism presented in this section applies also for small systems. The above treatment of the vanishing pore width provides such an application concretely. Moreover, the success in describing correctly the 3D to 2D crossover shows that our alternative approach possesses also some advantage compared to Hill's nanothermodynamics.

**Statistical mechanics**. The thermodynamic formalisms presented in the last section provide a macroscopic description. In this section, we present also a microscopic description based on statistical mechanics and discuss the connections between the two descriptions. For this, we consider a quite general model for a one-component fluid confined in a slit pore formed by two parallel impenetrable walls. The fluid particles interact with each other through a pair-wise additive potential, $u$, and with each wall

through $v_W$. The total interaction potential is given by,

$$\mathcal{U} = \sum_{i=1}^{N}\sum_{j>i}^{N}u(|\boldsymbol{r}_j - \boldsymbol{r}_i|) + \sum_{i=1}^{N}\left[v_W(z_i + L/2) + v_W(L/2 - z_i)\right], \qquad (37)$$

where $\boldsymbol{r}_i = (x_i, y_i, z_i)$ is the position vector of particle $i$ (see Fig. 1) and the interparticle potential considered in this subsection can be any generally used one. For the general discussion here, we just require that the fluid-wall interaction goes to infinity when a fluid particle is out the slit pore, i.e.,

$$v_W(z) = \infty, \qquad z < 0. \qquad (38)$$

Inside the pore, $v_W(z_i)$ can be a quite arbitrary finite potential.

Since the Helmholtz free energy is chosen as the thermodynamic potential for the presentation given in the Section "Thermodynamics", the canonical ensemble is considered for the corresponding presentation of statistical-mechanics results in this section. The Helmholtz free energy can be expressed either by,

$$\beta F(T, V, \mathcal{A}, N) = -\ln Z(T, V, \mathcal{A}, N), \qquad (39)$$

or

$$\beta\bar{F}(T, L, A, N) = -\ln\bar{Z}(T, L, A, N), \qquad (40)$$

where the partition function is respectively given by,

$$Z = \frac{1}{\Lambda^{3N}N!}\int_V\prod_{i=1}^{N}d\boldsymbol{r}_i e^{-\beta\mathcal{U}} = \frac{1}{\Lambda^{3N}N!}\prod_{i=1}^{N}\int_0^{\sqrt{A/2}}dx_i\int_0^{\sqrt{A/2}}dy_i\int_{-V/A}^{V/A}dz_i e^{-\beta\mathcal{U}}, \qquad (41)$$

$$\bar{Z} = \frac{1}{\Lambda^{3N}N!}\int_V\prod_{i=1}^{N}d\boldsymbol{r}_i e^{-\beta\mathcal{U}} = \frac{1}{\Lambda^{3N}N!}\prod_{i=1}^{N}\int_0^{\sqrt{A}}dx_i\int_0^{\sqrt{A}}dy_i\int_{-L/2}^{L/2}dz_i e^{-\beta\mathcal{U}}, \qquad (42)$$

and $\Lambda$ is the thermal wavelength.

*Normal pressure and a general contact-value theorem.* According to its definition given in Eq. (8), it is straightforward to obtain the statistical-mechanics expression of the normal pressure. To facilitate the calculation of the derivative, the change of variable, $z_i = L\hat{z}_i$ with $L = 2V/\mathcal{A}$, is made and the partition function becomes,

$$Z = \frac{(2V/\mathcal{A})^N}{\Lambda^{3N}N!}\prod_{i=1}^{N}\int_0^{\sqrt{A/2}}dx_i\int_0^{\sqrt{A/2}}dy_i\int_{-1/2}^{1/2}d\hat{z}_i e^{-\beta\mathcal{U}}. \qquad (43)$$

The final result for the normal pressure is given by,

$$\begin{aligned}
\beta p_{\perp} &= -\left[\frac{\partial(\beta F)}{\partial V}\right]_{T,N,\mathcal{A}} = -\frac{1}{A}\left[\frac{\partial(\beta F)}{\partial L}\right]_{T,N,\mathcal{A}} \\
&= \rho - \frac{\beta}{2L}\int dz_1\int d\boldsymbol{r}_{12}\rho^{(2)}(z_1, z_2, s_{12})u'(r_{12})\frac{(z_2 - z_1)^2}{r_{12}} \\
&\quad - \frac{\beta}{L}\int dz_1\rho(z_1)\left[v_W'(z_1 + L/2)\left(z_1 + \frac{L}{2}\right) + v_W'(L/2 - z_1)\left(\frac{L}{2} - z_1\right)\right],
\end{aligned} \qquad (44)$$

where $\boldsymbol{r}_{12} = \boldsymbol{r}_2 - \boldsymbol{r}_1$, $\rho = N(AL)^{-1}$, $\rho(z_1)$ is the density profile of the confined fluid, $\rho^{(2)}(z_1, z_2, s_{12})$, the two-body distribution function and,

$$s_{ij} = \sqrt{(x_j - x_i)^2 + (y_j - y_i)^2}. \qquad (45)$$

When the fluid interacts with the pore walls through a hard wall potential, i.e.,

$$v_W(z) = v_{HW}(z) = \begin{cases} \infty & , \quad z < 0 \\ 0 & , \quad z \geq 0 \end{cases}, \qquad (46)$$

the last term on the RHS of Eq. (44) vanishes.

For an inhomogeneous fluid in contact with a single hard wall, the well-known contact-value theorem (see e.g., the work of Henderson and Blum[22]) gives the exact result, $\beta p^{bulk} = \rho_c$, with

$p^{bulk}$ being the pressure in the bulk far from the wall and $\rho_c$ the fluid density at contact of the wall. For a fluid confined between two hard walls, it becomes highly inhomogeneous when the pore width is decreased, e.g., to a few particle diameters, and the fluid loses its bulk character in the pore. We show now, even in this case, a contact-value theorem can be established for a fluid confined between two hard walls. In this case, Eq. (44) becomes,

$$\beta p_\perp = \rho - \frac{\beta}{2L} \int dz_1 \int d\boldsymbol{r}_{12} \rho^{(2)}(z_1, z_2, s_{12}) u'(r_{12}) \frac{(z_2 - z_1)^2}{r_{12}}. \tag{47}$$

First, we need to rewrite the second term on the RHS of Eq. (47) with the help of the first equation of Born-Green-Yvon hierarchy, i.e.,

$$\frac{d\rho(z_1)}{dz_1} = -\beta\rho(z_1) \frac{dv^{ext}(z_1)}{dz_1} - \beta \int d\boldsymbol{r}_{12} \rho^{(2)}(z_1, z_2, s_{12}) u'(r_{12}) \frac{z_1 - z_2}{r_{12}}. \tag{48}$$

Multiplying the both sides of Eq. (48) by $z_1$ and integrating over $z_1$, we obtain,

$$\int dz_1 z_1 \frac{d\rho(z_1)}{dz_1} + \beta \int dz_1 z_1 \rho(z_1) \frac{dv^{ext}(z_1)}{dz_1}$$
$$= -\beta \int dz_1 \int d\boldsymbol{r}_{12} \rho^{(2)}(z_1, z_2, s_{12}) u'(r_{12}) \frac{(z_1 - z_2) z_1}{r_{12}} \tag{49}$$
$$= -\frac{\beta}{2} \int dz_1 \int d\boldsymbol{r}_{12} \rho^{(2)}(z_1, z_2, s_{12}) u'(r_{12}) \frac{(z_1 - z_2)^2}{r_{12}}.$$

When going to the last equality of Eq. 49, we used the symmetry relation, $\rho^{(2)}(\boldsymbol{r_1}, \boldsymbol{r_2}) = \rho^{(2)}(\boldsymbol{r_2}, \boldsymbol{r_1})$, which allows for replacing $(z_1 - z_2) z_1$ in the integrand by $(z_1 - z_2)^2/2$. Substituting Eq. (49) and $v^{ext}(z_1) = v_{HW}(z + L_z/2) - v_{HW}(L_z/2 - z)$ into Eq. (47), we obtain $\int_{-\infty}^{\infty} dz_1 z_1 \frac{d\rho(z_1)}{dz_1} = -\int_{-\infty}^{\infty} dz_1 \rho(z_1) = N/A$, $\rho(z_1) v'_{HW}(z + L/2) = \rho(z_1) e^{\beta v_{HW}(z+L/2)} \left[ e^{-\beta v_{HW}(z+L/2)} \right]'$ and finally,

$$\beta p_\perp = -\frac{2}{L} \int_{-\infty}^{\infty} dz_1 z_1 \rho(z_1) e^{\beta v_{HW}(z+L/2)} \delta\left(z + \frac{L}{2}\right)$$
$$= \rho\left(-\frac{L^+}{2}\right) = \rho\left(\frac{L^-}{2}\right). \tag{50}$$

where we have used the fact that the density profile is an even function. It is to be emphasized that the contact-value theorem given by Eq. (50) hold for any pore width. This exact result, which was already observed in the case of extreme confinement[23] but not proven, can serve as one accuracy check for a density functional theory dealing with fluids confined in a slit pore with hard walls. In the limit $L \to \infty$, we recover the contact-value theorem for a single wall since $p_\perp \to p^{bulk}$.

*Integrated transverse pressure.* The thermodynamic definition of the averaged traverse pressure is given by Eq. (13). Now, we find its statistical-mechanics expression again by calculating the corresponding derivative of Helmholtz energy. To facilitate this calculation, the change of variables, $x_i = \sqrt{A}\hat{x}_i$ and $y_i = \sqrt{A}\hat{y}_i$ are made and we obtain,

$$\beta\bar{\Sigma} = -\left[\frac{\partial(\beta\bar{F})}{\partial A}\right]_{T,L,N} = n - \frac{\beta}{4} \int dz_1 \int d\boldsymbol{r}_{12} \rho^{(2)}(z_1, z_2, s_{12})$$
$$u'(r_{12}) \left[\frac{(x_2 - x_1)^2}{r_{12}} + \frac{(y_2 - y_1)^2}{r_{12}}\right], \tag{51}$$

where $n = N/A$ is surface density.

*Surface tension.* We carry out also the calculation of the surface tension according to the thermodynamic definition given in Eq. (3). To facilitate this calculation, the change of variables, $x_i = \sqrt{A/2}\hat{x}_i$, $y_i = \sqrt{A/2}\hat{y}_i$ and $z_i = 2V\hat{z}_i/A$ are made and we obtain,

$$\beta\gamma = \beta\left[\frac{\partial F}{\partial A}\right]_{T,V,N} = \frac{\beta}{8} \int dz_1 \int d\boldsymbol{r}_{12} \rho^{(2)}(z_1, z_2, s_{12})$$
$$u'(r_{12}) \left[\frac{(x_2 - x_1)^2}{r_{12}} + \frac{(y_2 - y_1)^2}{r_{12}}\right]$$
$$- \frac{\beta}{4} \int dz_1 \int d\boldsymbol{r}_{12} \rho^{(2)}(z_1, z_2, s_{12}) u'(r_{12}) \frac{(z_2 - z_1)^2}{r_{12}}$$
$$- \frac{\beta}{2} \int dz_1 \rho(z_1) \left[v'_W(z_1 + L/2)\left(z_1 + \frac{L}{2}\right) + v'_W(L/2 - z_1)\left(\frac{L}{2} - z_1\right)\right]. \tag{52}$$

Again, for a pore with two hard walls, the last term on the RHS of Eq. (52) vanishes. The statistical-mechanics results obtained in Eqs. (44), (51) and (52) show that the thermodynamic relation, Eq. (18), holds perfectly. This provides the firm microscopic foundation of the thermodynamic formulations presented in Section Thermodynamics. One obvious advantage of the above thermodynamic route to the normal and averaged transverse pressures is that it avoids completely the uniqueness problem related to the choice of integration paths suffered by the route based on the mechanical definition of pressure tensor.

*Vanishing pore-width limit $L \to 0$ and dimensional crossover.* Now, we determine the limit values of the thermodynamic functions discussed in the last subsection when the pore width becomes vanishingly small. In this limit, we have,

$$\lim_{L_z \to 0} r_{12} = s_{12} = \sqrt{(x_2 - x_1)^2 + (y_2 - y_1)^2}, \tag{53}$$

$$\lim_{L_z \to 0} \rho^{(2)}(z_1, z_2, s_{12}) = \rho(z_1)\rho(z_2) g(0, 0, s_{12}) = \rho(z_1)\rho(z_2) g^{2D}(s_{12}). \tag{54}$$

Equation (54) shows clearly the decoupling of the normal and traverse variables. With the help of Eq. (54), we can readily show that the second term on the RHS of Eq. (47) vanishes when $L \to 0$. So, in this limit, the normal pressure is given by the equation of state of an ideal gas. Although $\rho \to \infty$, we find,

$$\lim_{L \to 0} \beta p_\perp L = n. \tag{55}$$

Substituting Eqs. (53) and (54) into Eq. (51), we can carry out the integration of the perpendicular variables and obtain,

$$\lim_{L \to 0} \beta\bar{\Sigma} = n - \frac{\beta}{4} \lim_{L \to 0} \int_{-L/2}^{L/2} dz_1 \rho(z_1) \int_{-L/2}^{L/2} dz_2 \rho(z_2) \int d\boldsymbol{s}_{12} s_{12} g^{2D}(s_{12}) u'(s_{12})$$
$$= n - \frac{\beta n^2}{4} \int d\boldsymbol{s}_{12} s_{12} g^{2D}(s_{12}) u'(s_{12}). \tag{56}$$

The right-hand-side (RHS) of Eq. (56) is nothing else but the expression of the pressure (times $\beta$) of a 2D fluid. This confirms the results, $\lim_{L \to 0}\bar{\Sigma} = p^{2D}$, obtained in the Subsection "Vanishing pore-width limit", i.e., Eq. (35). In a similar way, we find, from Eq. (52), the following result for the limit value of the surface tension,

$$\lim_{L \to 0} \beta\gamma = \frac{\beta n^2}{8} \int d\boldsymbol{s}_{12} s_{12} g^{2D}(s_{12}) u'(s_{12}). \tag{57}$$

In the vanishing pore width limit, the surface tension of the 3D confined fluid accounts for the non-ideal part of the pressure for

the limiting 2D fluid. The results given in Eqs. (55)–(57) show readily that the thermodynamic result of Eq. (30) holds.

The above results show how the dimensional crossover can be realized by confining a system with walls. Such a technique has been used for constructing density functionals which allow for accounting adequately dimensional crossover[24,25]. Now, we show that it is also possible to realize another dimensional crossover by adding hard walls in other directions to confine further the system. For the 2D to 1D crossover, the following fluid-wall interaction potential can be used,

$$\mathcal{V}_{ext} = \sum_{i=1}^{N} \Big[ v_{HW}(z_i + L/2) + v_{HW}(L/2 - z_i) \\ + v_{HW}(y_i + L_y/2) + v_{HW}(L_y/2 - y_i) \Big]. \tag{58}$$

where $L_y$ is the accessible distance between the two walls in $y$-direction. When $L \to 0$ and $L_y \to 0$, the system becomes a one-dimensional one. If a pair of hard walls is added also in $x$-direction, the crossover to 0D (just one fluid particle in the system) can be realized under the conditions $0 < L_x < \sigma$, $0 < L_y < \sigma$ and $0 < L < \sigma$ ($L_x$: accessible distance between the two walls in $x$-direction).

Before closing this subsection about the general statistical-mechanics results, we would like to point out also that all the above calculations can be also carried out in a different ensemble, e.g., a grand-canonical ensemble. When the pore width becomes small, the fluid structure near one wall is affected also by the presence of the other wall. Under this condition, $p_\perp \neq p^{bulk}$, their difference is Derjaguin's disjoining pressure[26,27] in a grand-canonical ensemble. It is recently revealed that a non-zero disjoining pressure is at the origin of distinct differential and integral surface tensions[13]. All the statistical-mechanics results presented above are valid also for any value of pore width, even when it becomes vanishingly small. One can wonder why the ensemble-dependence of thermodynamic results for small systems, pointed out by Hill, does not seem to show up here. From the very recent study of W. Dong[14], we know now that only integral intensive thermodynamic functions may manifest ensemble-dependence, e.g., the integral surface tension defined with grand potential can be different from that defined with isothermal-isobaric ensemble[14]. In the present work, we discuss only differential intensive thermodynamic functions and they are ensemble-independent.

**Illustration for a strongly confined hard-sphere fluid in a hard-slit pore.** In this section, we illustrate the thermodynamic quantities and their relations by using the statistical-mechanics results for a fluid of hard spheres (HS) of diameter $\sigma$. Despite its simplicity, many simulation results have shown that the HS model allows for describing many behaviors of real colloidal liquids. We consider a HS fluid confined in a slit pore formed by two hard walls with accessible width, $L$. Now, the specific inter-particle interaction is given by,

$$u^{HS}(|\mathbf{r}_i - \mathbf{r}_j|) = \begin{cases} \infty &, \quad |\mathbf{r}_i - \mathbf{r}_j| < \sigma \\ 0 &, \quad |\mathbf{r}_i - \mathbf{r}_j| \geq \sigma \end{cases}, \tag{59}$$

It is not possible to obtain exact analytical results for such a model for an arbitrary pore width, $L$. However, a previous work of two of us with S. Lang[15] has shown that a systematic analytical calculation of the free energy becomes possible for extreme confinement, i.e., $0 \leq L \leq \sigma$. In particular, it was proven that the thermodynamics of the confined HS fluid is identical to that of a 2D fluid of disks with a hard-core diameter $\sigma_L = \sqrt{\sigma^2 - L^2}$ and a

soft shell for $\sigma_L \leq |\mathbf{s}_i - \mathbf{s}_j| \leq \sigma$ [$\mathbf{s}_i = (x_i, y_i)$] where they interact via an effective potential interpolating between the infinite repulsion at $|\mathbf{s}_i - \mathbf{s}_j| = \sigma_L$ and the vanishing interaction at $|\mathbf{s}_i - \mathbf{s}_j| = \sigma$. A cluster expansion allows obtaining the following exact expression for the leading terms of the Helmholtz free energy[15],

$$\bar{F} = F_{id}^{3D} - F_{id}^{2D} + \widetilde{F}^{2D} + \Delta F + \mathcal{O}(nL^2)^2, \tag{60}$$

where $n = NA^{-1}$ is the 2D particle density, $F_{id}^{3D}$ and $F_{id}^{2D}$ are, respectively, the free energy of a 3D and 2D ideal gas, $\widetilde{F}^{2D}$ the free energy of a fluid of pure hard disks (HD) with a diameter $\sigma_L$, and,

$$\Delta F = \frac{5}{12}\pi k_B T N n L^2 g_+^{2D}(n\sigma^2) = \frac{5k_B T V^2}{6\sigma^2 A}(\beta p^{2D} - n), \tag{61}$$

is the leading-order correction originating from the coupling between the normal and transverse degrees of freedom. $g_+^{2D}(n\sigma^2) \equiv g^{2D}(\sigma^+; N, A)$ is the 2D radial distribution function at contact. In the second equality of Eq. (61), the following relation[28],

$$p^{2D} = k_B T n \left[ 1 + \frac{1}{2}\pi n \sigma^2 g_+^{2D}(n\sigma^2) \right], \tag{62}$$

was used. Replacing $g_+^{2D}(n\sigma^2)$ by the pressure $p^{2D}$ will facilitate the following discussions.

Now, we calculate various thermodynamic quantities up to the order of $nL^2$. From Eqs. (60), (61) and (10), we obtain readily the following result for the chemical potential,

$$\mu = \left(\frac{\partial F}{\partial N}\right)_{T,V,\mathcal{A}} = \widetilde{\mu}^{2D} - k_B T \ln\left(\frac{L}{\Lambda}\right) + \frac{5k_B T L^2}{6\sigma^2}\left[\beta\left(\frac{\partial p^{2D}}{\partial n}\right)_T - 1\right], \tag{63}$$

where $\widetilde{\mu}^{2D} = (\partial \widetilde{F}^{2D}/\partial N)_{T,V,A}$ is the chemical potential of a HD fluid of diameter $\sigma_L$. It is easy to check that $(\partial \bar{F}/\partial N)_{T,L,A}$ gives the same results as that given in Eq. (63), as it should be due to the findings in Section Statistical Mechanics. The normal pressure is given by,

$$p_\perp = -\left(\frac{\partial \bar{F}}{A \partial L}\right)_{T,A,N} = \frac{n k_B T}{L}\left[1 + \frac{1}{6}\pi n L^2 g_+^{2D}(n\sigma^2)\right] \\ = \frac{n k_B T}{L}\left[1 + \frac{1}{3}\left(\frac{L}{\sigma}\right)^2\left(\frac{\beta p^{2D}}{n} - 1\right)\right], \tag{64}$$

where we used the fact that $F_{id}^{2D}$ does not depend on $L$. In this calculation, one has to be careful for not setting $\left[\partial \widetilde{F}_{ex}^{2D}/(A\partial L)\right]_{T,A,N}$ to zero since $\widetilde{F}_{ex}^{2D}$ ($\widetilde{F}_{ex}^{2D} = \widetilde{F}^{2D} - F_{id}^{2D}$) depends on $n\sigma_L^2$ and $\sigma_L$ depends on $L$ (see the work of Franosch, Lang and Schilling[15]). The surface tension is given by,

$$\gamma = \left(\frac{\partial F}{\partial \mathcal{A}}\right)_{T,V,N} = \left(\frac{\partial \widetilde{F}^{2D}}{\partial \mathcal{A}}\right)_{T,V,N} + \frac{n k_B T}{2} \\ - \frac{5n k_B T L^2}{12\sigma^2}\left[\frac{\beta p^{2D}}{n} + \beta\left(\frac{\partial p^{2D}}{\partial n}\right)_T - 2\right]. \tag{65}$$

The derivative with respect to $\mathcal{A}$ accounts for both walls. Caution must be taken for calculating the first term on the RHS of Eq. (65), which is not equal to $-\widetilde{p}^{2D}/2 = \left[\partial \widetilde{F}^{2D}/\partial A\right]_{T,L,N}$ because $V$ is kept constant. As shown in the work of Franosch, Lang, and Schilling[15], $\widetilde{F}^{2D}$ depends not only on $A$ but also on $L$, via $\sigma_L$. For $T$

and $N$ constant, this implies,

$$d\widetilde{F}^{2D} = \left[\frac{\partial \widetilde{F}^{2D}}{\partial(2A)}\right]_{T,L,N} d(2A) + \left[\frac{\partial \widetilde{F}_{HD}^{2D}}{\partial L}\right]_{T,A,N} dL. \quad (66)$$

Then, Eq. (66) and $dV = AdL + LdA = 0$ lead to,

$$\left[\frac{\partial \widetilde{F}^{2D}}{\partial(2A)}\right]_{T,V,N} = \left[\frac{\partial \widetilde{F}^{2D}}{\partial(2A)}\right]_{T,L,N} - \frac{L}{2}\left(\frac{\partial \widetilde{F}^{2D}}{A\partial L}\right)_{T,A,N}$$
$$= -\frac{\widetilde{p}^{2D}}{2} + \frac{nk_BTL^2}{\sigma^2}\left(\frac{\beta p^{2D}}{n} - 1\right), \quad (67)$$

where the leading order term in the first equation in the work of Franosch, Lang and Schilling[15], i.e.,

$$-\left(\frac{\partial \widetilde{F}^{2D}}{A\partial L}\right)_{T,A,N} = -\left(\frac{\partial \widetilde{F}_{ex}^{2D}}{A\partial L}\right)_{T,A,N} = \pi n^2 k_B T L g_+^{2D}(n\sigma^2)$$
$$= \frac{2nk_BTL}{\sigma^2}\left(\frac{\beta p^{2D}}{n} - 1\right), \quad (68)$$

was used. Substituting Eq. (67) into Eq. (65) and using Eq. (62) lead to the following result for the surface tension,

$$\gamma = -\frac{1}{2}\left(\widetilde{p}^{2D} - k_BTn\right) + \frac{nk_BTL^2}{6\sigma^2}\left[\frac{\beta p^{2D}}{n} - 1\right]$$
$$+ \frac{5nk_BTL^2}{12\sigma^2}\left[\frac{\beta p^{2D}}{n} - \beta\left(\frac{\partial p^{2D}}{\partial n}\right)_T\right]. \quad (69)$$

Finally, we calculate the integrated transverse pressure. With Eq. (60), we obtain in leading order in $nL_z^2$,

$$\bar{\Sigma} = -\left(\frac{\partial F}{\partial A}\right)_{T,N,L} = \widetilde{p}^{2D} - \frac{5k_BTnL^2}{6\sigma^2}\left[\frac{\beta p^{2D}}{n} - \beta\left(\frac{\partial p^{2D}}{\partial n}\right)_T\right]. \quad (70)$$

We emphasized above that $-(\partial \widetilde{F}^{2D}/\partial \mathcal{A})_{T,V,N} \neq \widetilde{p}^{2D}/2$, but $-(\partial \widetilde{F}^{2D}/\partial A)_{T,L,N} = \widetilde{p}^{2D}$, which was used in deriving Eq. (70). This shows the importance of accounting properly for the condition under which the partial derivative is taken. The reader should note that the expression for $\bar{\Sigma}$ given in the work of Franosch, Lang, and Schilling[15] is not complete. The last term in Eq. (70) involving the derivative of $p^{2D}$ with respect to $n$, is missing since the dependence of $g_+^{2D}(n\sigma^2)$ on $A$ through $n$ was overlooked. This error has been corrected now[29]. The results for $p_\perp$, $\gamma$ and $\bar{\Sigma}$, given in Eqs. (64), (69) and (70), illustrate that Eq. (18) given in Section "Thermodynamics" holds perfectly up to the order of $nL^2$.

With the above results for the chemical potential, normal pressure, surface tension and integrated transverse pressure, one can readily check that the Euler relations given in Eqs. (22) and (25) indeed hold up to order $nL^2$. The same holds for the Gibbs-Duhem Eqs. (23) and (26). Concerning the 2D limit, Eqs. (63) and (70) show straightforwardly,

$$\lim_{L \to 0}\left[\mu + k_BT\ln\left(\frac{L}{\Lambda}\right)\right] = \mu^{2D}, \quad (71)$$

$$\lim_{L \to 0} \bar{\Sigma} = p^{2D}, \quad (72)$$

where $\mu^{2D} = \lim_{L\to 0}\bar{\mu}^{2D}$ and $p^{2D} = \lim_{L\to 0}\widetilde{p}^{2D}$ were used. They confirm the validity of the thermodynamic relations given by Eqs. (31) and (35) of Section "Thermodynamics". With the help of the results given in Eqs. (64) and (69), we find,

$$\lim_{L \to 0}\left(p_\perp L - 2\gamma\right) = nk_BT + \frac{\pi n^2 k_B T\sigma^2 g_+(n\sigma^2)}{2} = p^{2D}. \quad (73)$$

which corroborates Eq. (30) derived in Section "Thermodynamics".

## Conclusions

In the present contribution, we have explored the consequences of the different choices of the independent thermodynamic variables of the Helmholtz free energy for a confined fluid. For instance, for a one-component fluid in a slit pore composed of two parallel flat impenetrable walls with accessible width $L$ and wall area $A$, two alternative thermodynamic formulations exist, both are based on the Helmholtz free energy. They are denoted respectively as $F(T, V, \mathcal{A}, N)$ and $\bar{F}(T, L, A, N)$. They differ in the choice of one independent variable, i.e. either the volume $V = AL$ or the pore-width $L$, besides temperature, $T$, surface area, $A$ (or $\mathcal{A} = 2A$), and particle number, $N$. Consequently, the Euler relation (a consequence of the homogeneity of the free energy in its extensive variables), the Gibbs-Duhem equation and the equation of states, i.e., the derivatives of the free energy with respect to its independent variables, are different for $F$ and $\bar{F}$. Nevertheless, we showed that both formulations are completely equivalent, as well as how the equations of state of the two formulations are related to each other. This also leads to a more precise physical interpretation of the derivatives, $(\partial F/\partial \mathcal{A})_{T,V,N}$ and $(\partial \bar{F}/\partial A)_{T,L,N}$, which both have the dimension of a surface tension. While the former is the surface tension of the confined fluid, the latter gives the integrated transverse pressure.

Some general statistical-mechanics results valid for any pore width are also presented and they establish the connection of a microscopic description to thermodynamics. Our statistical-mechanics results provide a thermodynamic route for obtaining the respective microscopic expressions of the normal pressure, the integrated transverse pressure and the surface tension without resorting to the pressure tensor. Thus, the non-uniqueness problem of the pressure tensor is totally avoided. Moreove, a contact-value theorem is established for a fluid confined in a pore composed of two parallel hard walls with an arbitrary pore width.

Thermodynamics provides relations between different thermodynamic functions but does not give explicit expressions of thermodynamic potentials, $F$ and $\bar{F}$. In order to calculate the free energy, one has to resort to statistical mechanics. For a colloidal liquid of monodisperse hard spheres with diameter, $\sigma$, in a slit pore of accessible width $L$, it was proven that for $L \leq \sigma$, the Helmholtz free energy can be calculated analytically, taking $nL^2$ as a smallness parameter with $n = NA^{-1}$ being the 2D density[15]. With the result for the free energy given in the work of Franosch, Lang, and Schilling[15], we have calculated the equations of state and illustrated explicitly the consistency of the results obtained from the two alternative thermodynamic formulations. We point out that the relationship between the thermodynamic quantities derived from $F$ and $\bar{F}$ is not restricted to the strong confinement, i.e., $0 \leq L \leq \sigma$, but hold for all $L$ and for any one-component fluid in a slit geometry.

We also showed how the various thermodynamic quantities for strong confinement allow recovering those quantities of the corresponding 2D fluid in the vanishing pore-width limit, i.e., $L \to 0$. Since the particle density of the 3D fluid, $\rho = NV^{-1} = N(AL)^{-1}$, and the chemical potentials, $\mu$ and $\bar{\mu}$, diverge for $L \to 0$, it is not obvious that the 3D results converge properly to their 2D counterparts. Despite these divergences, we have shown in Sec. II.B how the corresponding thermodynamic quantities in 2D can be obtained by taking the limit $L \to 0$. In this limit, the 3D system shrinks to a vanishing size in one direction. It is remarkable to see that thermodynamics holds even for such a non-macroscopic system if the divergences in the vanishing pore-width limit are treated properly. Recently, one of us has proposed an approach[13,14] alternative to Hill's nanothermodynamics by emphasizing the importance to account adequately for the surface contribution to thermodynamic potentials without resorting to

Hill's replica trick. The results of the present work provide support to this new approach for elaborating thermodynamics of small systems.

## Data availability

Data sharing not applicable to this article as no datasets were generated or analysed during the current study.

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

## Acknowledgements

R.S. would like to thank Jörg Baschnagel for helpful discussions and bringing the work of Varnik[17] to our attention and acknowledges that the present work was partially funded by the Deutsche Forschungsgemeinschaft (DFG, German Research Foundation)-SFB TRR 146, Project No. 233630050. W.D. is grateful for the financial support from Joint Institute for Science and Society (JoRISS) of Ecole Normale Supérieure de Lyon and East China Normal University, as well as that from China Hunan Provincial Science and Technology Department (project No 2019RS1031). T.F. gratefully acknowledges the support by the Austrian Science Fund (FWF): I 5257-N.

## Author contributions

All the authors, W.D. T.F. and R.S., participate the conception and investigation of the project, the analysis of the results, the writing and editing of the manuscript.

## Competing interests

The authors declare no competing interests.
