## [Peer review file · Communications Physics]

Reviewers' comments:

Reviewer #1 (Remarks to the Author):

The authors present an elegant analysis of the thermodynamics of fluids confined between parallel impenetrable walls. They show two different, but ultimately equivalent, formulations of the free energy in terms of independent variables. This development clarifies the surface tension and its relationship to the surface pressure of Langmuir films. It also reveals the relationship between the thermodynamics of the confined 3D fluid and the 2D fluid in the limit of vanishing pore width. This presentation stands out as the clearest that I have seen on this topic. They further explore these implications of these relationships using a fully analytical treatment of the statistical mechanics of hard spheres under extreme confinement, using a truncated cluster expansion.

This paper is an important contribution, which I suspect will be widely appreciated by the statistical mechanics community. However, the authors have the opportunity to increase the impact of the paper by commenting on how this development may inform (or relate to) the development of classical density functional theories (DFTs) of inhomogeneous fluids. In those systems, it was similarly realized that the exact excess free energy functional of the average one-body density could be related to a lower-dimensional version by a confinement-induced shrinking of the former's dimensionality. This dimensional crossover idea introduced logical consistency checks in the development of such theories (see, e.g., Rosenfeld et al., *J. Phys.: Condens. Matter* 8 (1996) L577–L581 and references therein). If the authors could clarify how the current development connects with the earlier work on DFT, it would help to contextualize the implications of their advancement.

It might also be interesting for the authors to translate their results for hard spheres in extreme confinement into the language of statistical geometry. Formal statistical geometric relationships for the pressure and the excess chemical potential of the hard-sphere fluid, in terms of averages of the available volume/surface area, are known for bulk hard-sphere fluids in any spatial dimension (see, e.g., Sastry et al., *Mol. Phys.* 95:2, 289-297 Corti and Bowles, *Mol. Physics*, 96:11, 1623-1635, 1999). It would be interesting to consider how the author's results for extreme confinement relate to the dimensional crossover from three-dimensional thermodynamics and available volumes/surface areas to their two-dimensional counterparts as the pore width shrinks to zero.

Reviewer #2 (Remarks to the Author):

In this paper, the authors study a thermodynamic theory of strongly confined fluids in the limits of vanishing pore width, which actually means how to develop the thermodynamics of a fluid to study the 2D case as a limit of a 3D case when one of the dimensions tends to zero.

The theory was first developed by some of the authors in *PhysRevLett* 109, 240601 (2012), corrected for some technical errors and finally amended after comments by another of the authors with the modifications published as an erratum (see *PhysRevLett* 128. 209902(E) 2022).

The present manuscript is a consolidation of the whole story in a single paper. After reading it in detail, I do not see any new material from the previously published articles mentioned above. It is true that the new article consolidates the different technical errors detected along the years and also presents the formalism in a more detailed way. The present manuscript has some pedagogical value in the sense that allows readers to see a consolidated version of all the material without the need to check the different corrections to know which expression is still valid or not, but the manuscript itself contains no new science. The reason for a new publication in a primary journal is therefore not justified, so I do not agree with publications of this manuscript in its present form. Maybe some sort of secondary publication such a review, perspective or similar could be considered if there is enough material for that.

In addition to that, in the whole theory I miss a more detailed discussion of the microscopic fundamentals. The theory is postulated from macroscopic arguments (which is OK) but a detailed connection of the fundamental hypothesis with the microscopic theory (not just for the application to hard spheres developed right now) is highly desirable. In this sense, the classical book "molecular theory of capillarity" by Rowlinson and Widom may serve as a guide. I think that further work in this theory requires a substantial development of its microscopic foundations.

I also have a couple of minor points.

The authors employ in eqs.(19)-(20) the so-called mechanical definition for the surface tension based on the local pressure tensor. The definition for the local pressure tensor is non-unique, a fact which has generated a lot of debate from the conceptual point of view and generates a lot of complications from the practical side of calculating this function in molecular modeling of fluids (see for example E.R. Smith (2022), *Molecular Simulation*, 48:1, 57-72, DOI: 10.1080/08927022.2021.1953697) . I think that a discussion here is needed, including a comment on whether the expressions are affected or not by different choices of the definitions for the local pressure tensor.

The authors mention Hill's theory for the thermodynamics of small systems in discussing the motivation for their work. In fact, Hill shows several complex issues arising in the thermodynamics of small systems such as non-equivalence of different ensembles, and "non extensivity" of different thermodynamic potentials with some "unusual" exponents in the dependence of thermodynamic potentials from extensive variables. I wonder how all these results from Hill formalism relate to the formalism presented here which is more closer to classical macroscopic thermodynamics than the theory of thermodynamics of small systems. Is that because the system considered here is not really a small system (in the sense that it is a system in the thermodynamic limit but going from 3D to 2D?)

Reviewer #3 (Remarks to the Author):

Please see attached file.

In this work, Dong et al. have come up with an extended form of Helmholtz free energy of 3D confined fluids, which can be written as the function of different sets of intensive and extensive parameters. They further showed that this thermodynamic formalism can approach to that of 2D confined fluids when the distances between two slabs approaches zero. This thermodynamic formalism seems to be interesting and may be useful for studying the properties of confined liquids. I suggest the authors considering the following points when revising the manuscript.

1. The authors have shown two forms of the Helmholtz free energy for the confined liquids. The authors did not make it very clear how they are connected. In the introduction part of the manuscript, the authors have mentioned Legendre transforms. However, after finish reading this manuscript, I didn't see F and \bar{F} are connected via Legendre transform, and they seem to be equivalent to each other as the authors indicated. How are they equivalent to each other and what is the motivation for using different set of parameters to describe the same thermodynamic potential? Moreover, since the Helmholtz free energy is the Legendre transform of the energy function: $F = U - TS$, can both F and \bar{F} be derived from the same energy function?
2. During the test of their free energy model, the authors have shown that the intensive parameters such as p and γ can approach to 2D limit for the HSHW system. I also notice that this work depends heavily on the work in Ref. 13. Ref. 13 has also studied the phase transition in HSHW liquids. Does the free energy formalism presented in this work also applicable to phase transition?
3. I recommend the authors discussing how their model can be connected with molecular simulation results of confined fluid, since the hard sphere system tested in this work is a highly idealized system.

Point-by-point response to the reviewers' comments

We considered carefully all the comments and suggestions of all reviewers. A point-by-point response is given below to address them. Reviewers' comments are reproduced in green and separated in paragraphs to facilitate the point-by-point response.

Reviewer #1:

1-1) The authors present an elegant analysis of the thermodynamics of fluids confined between parallel impenetrable walls. They show two different, but ultimately equivalent, formulations of the free energy in terms of independent variables. This development clarifies the surface tension and its relationship to the surface pressure of Langmuir films. It also reveals the relationship between the thermodynamics of the confined 3D fluid and the 2D fluid in the limit of vanishing pore width. This presentation stands out as the clearest that I have seen on this topic. They further explore these implications of these relationships using a fully analytical treatment of the statistical mechanics of hard spheres under extreme confinement, using a truncated cluster expansion.

Reply: We are really grateful to your high praises to our work reported in this manuscript.

1-2) This paper is an important contribution, which I suspect will be widely appreciated by the statistical mechanics community. However, the authors have the opportunity to increase the impact of the paper by commenting on how this development may inform (or relate to) the development of classical density functional theories (DFTs) of inhomogeneous fluids. In those systems, it was similarly realized that the exact excess free energy functional of the average one-body density could be related to a lower-dimensional version by a confinement-induced shrinking of the former's dimensionality. This dimensional crossover idea introduced logical consistency checks in the development of such theories (see, e.g., Rosenfeld et al., J. Phys.: Condens. Matter 8 (1996) L577–L581 and references therein). If the authors could clarify how the current development connects with the earlier work on DFT, it would help to contextualize the implications of their advancement.

Reply: Your comment indicates an interesting issue. What we have achieved for the 3D to 2D crossover is made within a pure thermodynamic framework in Sec. II. Thus, the inhomogeneity of the system is not treated explicitly, but it is accounted for only through the surface contribution to the thermodynamic potential. In the revised manuscript, we added a totally new section, Sec. III, to present some general statistical-mechanics results, which may be useful for the issue you indicated here. For example, the general contact-value theorem established for a slit pore with an arbitrary pore width between the hard walls provides an exact result that any accurate density functional theory should satisfy (please see lines 307-330 in blue in the revised manuscript). In the new Sec. III, we added also a paragraph to discuss possibility to consider more general dimensional cross-overs by introducing additional walls (please see lines 372-381 in blue in the revised manuscript).

1-3) It might also be interesting for the authors to translate their results for hard spheres in extreme confinement into the language of statistical geometry. Formal statistical geometric relationships for the pressure and the excess chemical potential of the hard-sphere fluid, in terms of averages of the available volume/surface area, are known for bulk hard-sphere fluids in any spatial dimension (see, e.g., Sastry et al., Mol. Phys. 95:2, 289-297 Corti and Bowles, Mol. Physics, 96:11, 1623-1635, 1999).

It would be interesting to consider how the author's results for extreme confinement relate to the dimensional crossover from three-dimensional thermodynamics and available volumes/surface areas to their two-dimensional counterparts as the pore width shrinks to zero.

Reply: Thank you for suggesting also this interesting issue, which is certainly a perspective that our present work can contribute to develop. To the best of our knowledge, the statistical-geometry results for hard spheres in different dimensions have been obtained only for the bulk system in each corresponding dimension. Investigating the 3D to 2D crossover of the statistical geometry requires extending it to the inhomogeneous confining situation. Such knowledge is currently lacking. So, we do not see how the translation between our present work and the statistical geometry can be accomplished straightforwardly. But, this opens certainly some very nice perspectives for future investigations.

Reviewer #2:

2-1) In this paper, the authors study a thermodynamic theory of strongly confined fluids in the limits of vanishing pore width, which actually means how to develop the thermodynamics of a fluid to study the 2D case as a limit of a 3D case when one of the dimensions tends to zero. The theory was first developed by some of the authors in *PhysRevLett* 109, 240601 (2012), corrected for some technical errors and finally amended after comments by another of the authors with the modifications published as an erratum (see *PhysRevLett* 128. 209902(E) 2022).

The present manuscript is a consolidation of the whole story in a single paper. After reading it in detail, I do not see any new material from the previously published articles mentioned above. It is true that the new article consolidates the different technical errors detected along the years and also presents the formalism in a more detailed way. The present manuscript has some pedagogical value in the sense that allows readers to see a consolidated version of all the material without the need to check the different corrections to know which expression is still valid or not, but the manuscript itself contains no new science. The reason for a new publication in a primary journal is therefore not justified, so I do not agree with publications of this manuscript in its present form. Maybe some sort of secondary publication such a review, perspective or similar could be considered if there is enough material for that.

Reply: The work published in *PRL* 109, 240601 (2012), by two of us and S. Lang, presented a statistical-mechanics method for calculating exactly and analytically the Helmholtz free energy of a hard-sphere fluid confined in a very narrow hard slit pore ($L/\sigma < 1$) to the leading order of L/σ . Nor in the original *PRL* paper neither in the two subsequent errata, one can find the main results of the present work, i.e., the general thermodynamic formalism, the 3D to 2D dimensional crossover and their corroboration by exact statistical-mechanics results. So, our present manuscript is neither a long version of the original *PRL* nor a compilation of the previously published errata. In addition, we would like to draw Reviewer #2's attention about the fact that our thermodynamic formalism, as well as all results in the new section III hold for all pore widths L , in contrast to the more restricted results of Ref.[15] (Ref.[13] in the first version of our manuscript). This point is now emphasized more explicit in the revised manuscript (please see lines 203 and 478-479 in blue in the revised manuscript).

When the surface contribution becomes important in the thermodynamic potential of a system, the system is highly inhomogeneous and becomes anisotropic in the surface region. It is well-known that the thermodynamics of macroscopic systems does not treat such anisotropy. For example, one can change the volume of a macroscopic system in any way and obtains the same modification of the thermodynamic potential. However, this is no longer true for an inhomogeneous and anisotropic system. The well-known example is that the pressure is no longer a scalar but becomes a second-order tensor. Changing system's volume along the direction normal or parallel to the surface does not result in the same change of the thermodynamic potential. So, there is a clear need to extend the thermodynamics of macroscopic systems for dealing with appropriately these specificities related to inhomogeneity and anisotropy. To the best of our knowledge, such a thermodynamic formalism is still lacking. Our present work aims precisely at filling this gap. We show that the two different but equivalent thermodynamic descriptions come precisely from the system's anisotropy. The pressure tensor was defined from mechanics consideration. From our formalisms, we find also how to obtain the averaged normal and transverse pressures from thermodynamics and this is a totally original result. All these subjects have not been published in the previous PRL papers you mentioned above. We believe all these are the "new science" offered by the present work. In the revised manuscript, a list of open questions investigated and answered in the present work is given in the introduction (please see lines 73-79 in blue in the revised manuscript).

2-2) In addition to that, in the whole theory I miss a more detailed discussion of the microscopic fundamentals. The theory is postulated from macroscopic arguments (which is OK) but a detailed connection of the fundamental hypothesis with the microscopic theory (not just for the application to hard spheres developed right now) is highly desirable. In this sense, the classical book "molecular theory of capillarity" by Rowlinson and Widom may serve as a guide. I think that further work in this theory requires a substantial development of its microscopic foundations.

Reply: We are grateful for this useful suggestion. Now, a totally new section, Sec. III, is added in the revised manuscript to present the general statistical-mechanics results corresponding to the main thermodynamic functions considered in this work, as well as the proof of a general contact-value theorem (please see lines 268-394 in blue in the revised manuscript). The more specific results for a HS fluid are now put in Sec. IV. We hope that the new Sec. III gives the microscopic fundamentals you wish to see. The general statistical-mechanics results are perfectly consistent with the thermodynamic ones given in Sec. II.

2-3) I also have a couple of minor points.

The authors employ in eqs.(19)-(20) the so-called mechanical definition for the surface tension based on the local pressure tensor. The definition for the local pressure tensor is non-unique, a fact which has generated a lot of debate from the conceptual point of view and generates a lot of complications from the practical side of calculating this function in molecular modeling of fluids (see for example E.R. Smith (2022), *Molecular Simulation*, 48:1, 57-72, DOI: 10.1080/08927022.2021.1953697). I think that a discussion here is needed, including a comment on whether the expressions are affected or not by different choices of the definitions for the local pressure tensor.

Reply: You raised here an important question about the non-uniqueness problem related to the mechanical definition of the pressure tensor. Our eq.(18) is obtained purely from thermodynamics but not from the mechanical definition of the pressure tensor. The comparison of eq.(18) with the surface tension expressed in terms of pressure tensor's components is just made to show that $\bar{\Sigma}$ is related to the integrated transverse component of the pressure tensor. The thermodynamic definition of $\bar{\Sigma}$ is given by eq.(13) which does not resort to any choice of the integration paths causing the non-uniqueness problem with the mechanical definition of the pressure tensor. So, to your question: “whether the expressions are affected or not by different choices of the definitions for the local pressure tensor.”, we answer clearly no and our thermodynamics formalism does not suffer from the non-uniqueness problem met with the mechanical definition of the pressure tensor. This highlights one advantage of our approach. Moreover, this is concretely illustrated with the general statistical-mechanical results added in the new Sec. III (please see lines 349-351 in blue in the revised manuscript). In this respect, our results here contribute to settle a long debate: i.e., no problem of uniqueness with the thermodynamic route to surface tension.

2-4) The authors mention Hill's theory for the thermodynamics of small systems in discussing the motivation for their work. In fact, Hill shows several complex issues arising in the thermodynamics of small systems such as non-equivalence of different ensembles, and “non extensivity” of different thermodynamic potentials with some “unusual” exponents in the dependence of thermodynamic potentials from extensive variables. I wonder how all these results from Hill formalism relate to the formalism presented here which is more closer to classical macroscopic thermodynamics than the theory of thermodynamics of small systems. Is that because the system considered here is not really a small system (in the sense that it is a system in the thermodynamic limit but going from 3D to 2D?)

Reply: You are perfectly right. Our approach looks much closer to Gibbs' thermodynamics of interfaces than to Hill's nanothermodynamics. A recent work of one of us (ref. [13] in the revised manuscript which was ref.[19] in the previous version) has shown that Hill's approach is not the unique one for describing the thermodynamics of small systems and that Gibbs' thermodynamics of interfaces can be extended to describe the thermodynamics of small systems and to reveal all the particular behaviors of small systems. Despite revealing distinct differential and integral pressures, ref.[13] revealed also the existence of distinct differential and integral surface tensions. A recent work of W. Dong (just accepted for publication by Nature Communications, cited in the revised manuscript as ref.[14]) shows that the non-equivalence of different ensembles concerns only the integral intensive variables but not the differential intensive variables (i.e., those derived from the derivative of a thermodynamic potential with respect to an extensive variable). In the present manuscript, we discuss only differential intensive variables, i.e., differential pressure, differential chemical potential and differential surface tension. So, the ensemble-dependence is not concerned, so not discussed. Moreover, it is straightforward to check that all the general statistical-mechanics results in Sec. III derived with canonical ensemble remain the same if a grand-canonical ensemble or a pTN -ensemble is used. This is explicitly pointed out in the revised manuscript (please see 382-394 lines in blue in the revised manuscript). Nevertheless, your remarks on this topic called our attention to explain more clearly that the thermodynamic formalisms presented in this work is perfectly capable of revealing all the characteristic behaviors manifested in small systems. We added a paragraph to address your questioning: “I wonder how all these results from Hill formalism relate to the formalism presented

here which is more closer to classical macroscopic thermodynamics than the theory of thermodynamics of small systems.” (please see lines 258-267 in blue in the revised manuscript). Moreover, it is to be pointed out that all the thermodynamic quantities considered in our work are in principle measurable while the subdivision potential associated with Hill’s replica trick does not seem to be measurable. Anyway, no experimental measurement for it has ever been reported.

To your question: “Is that because the system considered here is not really a small system (in the sense that it is a system in the thermodynamic limit but going from 3D to 2D?)”, we give the following clear answer. The system considered in our present work can be extremely small in one direction when we consider the limit $L \rightarrow 0$. To the best of our knowledge, nobody has shown that thermodynamics still holds under such a condition although the 3D to 2D dimensional crossover is a physically appealing idea (please see again lines 258-267 in blue in the revised manuscript). When the pore width becomes very small, the surface region invades actually the whole system and the surface contribution to the thermodynamic potential becomes predominant. This is one main basic characteristic of any small system (please see [13] and [14] to be published soon).

Reviewer #3:

3-1) In this work, Dong et al. have come up with an extended form of Helmholtz free energy of 3D confined fluids, which can be written as the function of different sets of intensive and extensive parameters. They further showed that this thermodynamic formalism can approach to that of 2D confined fluids when the distances between two slabs approaches zero. This thermodynamic formalism seems to be interesting and may be useful for studying the properties of confined liquids. I suggest the authors considering the following points when revising the manuscript.

Reply: Thank you for your positive evaluation of our work with relevant comments and questions which are considered carefully below.

3-2) 1. The authors have shown two forms of the Helmholtz free energy for the confined liquids. The authors did not make it very clear how they are connected. In the introduction part of the manuscript, the authors have mentioned Legendre transforms. However, after finish reading this manuscript, I didn't see F and \bar{F} are connected via Legendre transform, and they seem to be equivalent to each other as the authors indicated. How are they equivalent to each other and what is the motivation for using different set of parameters to describe the same thermodynamic potential? Moreover, since the Helmholtz free energy is the Legendre transform of the energy function: $F = U - TS$, can both F and \bar{F} be derived from the same energy function?

Reply: In the introduction, we added a few lines to dissipate the confusion with the change of independent variables through a Legendre transform and the possible different choices for a same thermodynamic potential (please see lines 49-53 in blue in the revised manuscript). Concerning your question: “How are they equivalent to each other?”, please see lines 131-132, 189-192 (in blue) of the revised manuscript. To answer your question: “what is the motivation for using different set of parameters to describe the same thermodynamic potential?”, it is to note the choice of different sets of independent variables follows the general spirit of this matter in thermodynamics. One set can be more convenient for some experimental or simulation situations while another set is more suitable in other situations (please see lines 52-53 in blue in the revised manuscript). Moreover, the two equivalent formalisms result in different intensive variable and different Gibbs-Duhem equations. So,

it is worthwhile to work them out explicitly. The answer is yes to your question: “can both F and \bar{F} be derived from the same energy function?”. It is to note that there are two possible choices of independent variables for the internal energy, $U(S, V, A, N)$ and $\bar{U}(\bar{S}, L, A, N)$, and for entropy, $S(U, V, A, N)$ and $\bar{S}(\bar{U}, L, A, N)$. Legendre transform gives respectively, $F = U - TS$ and $\bar{F} = \bar{U} - T\bar{S}$.

3-3) 2. During the test of their free energy model, the authors have shown that the intensive parameters such as p and γ can approach to 2D limit for the HSHW system. I also notice that this work depends heavily on the work in Ref. 13. Ref. 13 has also studied the phase transition in HSHW liquids. Does the free energy formalism presented in this work also applicable to phase transition?

Reply: We suppose that by “free energy model” you mean the thermodynamics formalism presented in Sec. II.A. This formalism is quite general, not only valid for a HSHW system but also for any fluid confined in a slit pore with hard walls or with more general fluid-wall interaction potentials. However, for studying the 3D to 2D dimensional crossover, we use a slit pore with two hard walls in Sec. II.B. The validity of the results in Sec. II.B is not limited to a HS fluid and the fluid can have an arbitrary interparticle interaction potential. Concerning the connection between the present work and the previous one of two of us with S. Lang ([13] in the previous version and [15] in the revised version), we use some previous results [15] for illustration given in Sec. IV. Even this part contains some new results, e.g., the expressions for the chemical potential, for the surface tension, consistency check with Gibbs-Duhem equation, the vanishing pore limit given in eqs. (71)-(73). All these results and those in the other parts of the manuscript are original ones, have never been published before.

The answer is yes to your question: “Does the free energy formalism presented in this work also applicable to phase transition?”. This affirmation is added in the revised manuscript (please see lines 199-202 in blue in the revised manuscript).

3-4) 3. I recommend the authors discussing how their model can be connected with molecular simulation results of confined fluid, since the hard sphere system tested in this work is a highly idealized system.

Reply: Although the hard sphere system is an idealized model, many simulation results have shown that it allows for describing many behaviors of real colloidal liquids. This fact is explicitly indicated in the revised manuscript (please see lines 398-399 in blue in the revised manuscript).

In summary, we hope the above point-by-point response clarifies all the important concerns raised by the reviewers. The revised manuscript is an improved and substantially enriched one, which reaches the high standards of the reviewers.

REVIEWERS' COMMENTS:

Reviewer #1 (Remarks to the Author):

The authors have adequately addressed all reviewer concerns, and the revised paper is suitable for publication.

Reviewer #3 (Remarks to the Author):

The author has properly addressed all the issues raised by the reviewers, and this manuscript can be accepted for publication.